# An Observational Case-Control Study to Determine Human Immunodeficiency Virus and Host Factor Influence on Biomarker Distribution and Serodiagnostic Potential in Adult Pulmonary Tuberculosis

**DOI:** 10.3390/tropicalmed4020057

**Published:** 2019-03-31

**Authors:** Khayriyyah Mohd Hanafiah, Mary Louise Garcia, David Andrew Anderson

**Affiliations:** 1Life Sciences, Macfarlane Burnet Institute, Melbourne 3004, Victoria, Australia; Mary.garcia@burnet.edu.au (M.L.G.); David.anderson@burnet.edu.au (D.A.A.); 2School of Biological Sciences, Universiti Sains Malaysia, Pulau Pinang 11600, Malaysia; 3Department of Immunology, Nursing and Health Sciences, Faculty of Medicine, Monash University, Clayton 3800, Victoria, Australia

**Keywords:** biomarkers, C-reactive protein, antibody, enzyme-linked immunosorbent assay, tuberculosis, HIV infections

## Abstract

Influence of host factors, including human immunodeficiency virus (HIV) co-infection, on the distribution and diagnostic potential of previously evaluated biomarkers of pulmonary tuberculosis (PTB), such as anti-antigen 60 (A60) immunoglobulin (Ig) G, anti-A60 IgA, and C-reactive protein (CRP), remain unclear. Anti-A60 IgG, anti-A60 IgA, and CRP in PTB and non-PTB patient sera (*n* = 404, including smear-positive/negative, culture-positive (SPCP/SNCP) and HIV+ve/−ve) were measured by enzyme-linked immunoassay and statistically analysed. In multinomial logistic regression, expectoration, chest pain, wasting, and culture count positively associated with CRP (*p* < 0.001), while smear count positively associated with anti-A60 IgG (*p* = 0.090). Expectoration and enlarged lymph nodes negatively associated with anti-A60 IgA (*p* = 0.018). Biomarker distribution and diagnostic potential varied significantly by symptoms and bacilli burden, and across different PTB subpopulations. CRP was correlated poorly with anti-A60 antibodies, while anti-A60 IgA and IgG were correlated in non-tuberculosis (TB) and SPCP patients (*p* < 0.001). When combined, anti-A60 IgG and CRP best discriminated SPCP/HIV−ve from non-TB (AUC: 0.838, 95% CI: 0.783–0.894), while anti-A60 IgA and CRP performed best in discriminating HIV+ve PTB from non-TB (AUC: 0.687, 95% CI: 0.598–0.777). Combined CRP and anti-A60 antibodies had significantly reduced accuracy in SNCP and SNCP/HIV+ve compared to SPCP/HIV−ve subpopulations. The complex relationships between host factors and biomarkers suggest their limited utility, especially in SNCP/HIV+ve subpopulations, highlighting the importance of examining host response and immune biomarkers across relevant patient subpopulations.

## 1. Introduction

Pulmonary tuberculosis (PTB) remains an important cause of morbidity and mortality, particularly in low-middle income countries [1] and areas with high prevalence of human immunodeficiency virus (HIV) [2]. Public health strategies, such as the direct observed therapy shortcourse (DOTS), focus on diagnosis and treatment of active PTB cases as the primary source of ongoing transmission events [3]. However, case detection of active PTB relies on sputum-based diagnostics—either through detection of acid-fast bacilli (AFB) under smear microscopy or culture, or detection of *Mycobacterium tuberculosis* (MTB) nucleic acid—the use of which are generally confined to laboratories, and do not reach the majority of people at risk for PTB [4,5]. These challenges are compounded in subpopulations of PTB, such as HIV-positive patients, who also have higher mortality due to difficulty and subsequent delay in diagnosis [6,7]. Recently, an accurate biomarker-based non-sputum assay independent of HIV status and other host factors have been highlighted as a high-priority target product for tuberculosis (TB) diagnostic research [8,9]. As serodiagnostic assays are more amenable to point-of-care (POC) translation [10], detection of blood-based biomarkers remains an attractive approach. However, research and product development of biomarker-based assays appear to have stalled [1,11], in part due to scientific challenges arising from heterogenous patient immune response and the complex spectrum of TB disease manifestation [12].

Antibodies against antigen 60 (A60) and levels of the acute-phase C-reactive protein are among biomarkers commonly evaluated for serological diagnosis of PTB. Anti-A60 antibodies initially showed good diagnostic potential, but subsequently demonstrated variable accuracy in areas with high PTB prevalence, and among HIV-positive patient subpopulations [13,14,15]. C-reactive protein has shown promising results, particularly in HIV-positive individuals, but it is not PTB-specific and is elevated in several inflammatory conditions [16,17]. Furthermore, factors such as host immunogenetic background [18], AFB burden and stage of disease [19,20] have been hypothesized to influence the heterogeneous pattern of immune response, including dominance of different immunoglobulin isotypes [21,22] typically observed in confirmed PTB and patients for whom TB was provisionally diagnosed but later ruled out [23,24]. The aim of this study was to evaluate distribution of C-reactive protein and anti-A60 immunoglobulin G (IgG) and immunoglobulin A (IgA) in different PTB diagnostic groups by HIV status, determine variation in diagnostic performance and identify the host and clinical characteristics that may influence the distribution of these biomarkers to inform PTB diagnostic biomarker research.

## 2. Materials and Methods

### 2.1. Study Population

Serum samples (*n* = 404) from Viet Nam and South Africa collected from July 2009 to December 2012 from patients provisionally diagnosed with PTB, then confirmed using solid or liquid TB culture (genotype information unavailable) and concentrated Ziehl-Neelsen microscopy, were acquired from repositories managed by the Foundation for Innovative New Diagnostics (FIND) TB [25]. Samples were collected at WHO-approved centres from eligible consenting adults (>18 years) presenting with cough ⩾3 weeks’ duration, pre-treatment, and were processed without identifying information [26]. Active PTB (*n* = 222) subgroups include HIV-positive (HIV+ve) smear-negative culture-positive (SNCP) (*n* = 50), HIV+ve smear-positive culture-positive (SPCP) (*n* = 49), HIV-negative (HIV−ve) SNCP (*n* = 50), and HIV−ve SPCP (*n* = 73), while non-TB (*n* = 182) subgroups include HIV+ve non-TB (*n* = 50) and HIV−ve non-TB patients (*n* = 132). Non-TB patients tested negative for smear and culture at enrolment and at two months follow-up, although latent and extrapulmonary TB (LTB, ETB) could not be excluded. Available chest x-ray conclusions for 82 non-TB patients (45%) indicate that 46 non-TB patients were diagnosed with “pneumonia or atypical TB”, seven were diagnosed as “TB likely”, two patients were diagnosed with chronic obstructive pulmonary disorder (COPD), and one patient was diagnosed with Pneumocystis pneumonia (PCP), while nine patients were noted to have “evidence of previous TB”. In this observational case-control design, the PTB subgroup sample sizes are estimated to be able to detect a difference of 0.10, from AUC 0.78 to 0.90 with 95% confidence and 80% power [27]. Samples were stored frozen at −20 °C prior to use.

Experiments involving human sera samples were conducted with approval from the Alfred Ethics Committee (Approval Number: 169/13). 

### 2.2. Enzyme-Linked Immunoassay (ELISA)

Anti-A60 IgG and IgA were measured using an in-house ELISA protocol. Microtiter plates (Greiner Bio-One, Frickenhausen, Germany) were coated with 7 μg/mL of solubilised A60 complex (PBC Maes, Strasbourg, France) and blocked. 100 μL/well of serum sample diluted 1:100 in diluent comprising 1% (wt./vol) Bovine Serum Albumin (BSA) (Sigma-Aldrich, St. Louis, MO) and 0.05% (wt./vol) Tween-20 (Amresco, Solon, OH) in Phosphate Buffered Saline (PBS) (Oxoid, Hampshire, England) were added in duplicates. Plates were incubated at 37 °C for one hour, and washed thrice by dispensing and aspirating 250 μL/well of 0.9% (wt./vol) NaCl then blotted dry between each step. 100 μL/well of either horseradish peroxidase (HRP)-labelled rabbit anti-human IgG (Dako, Glostrup, Denmark), or HRP-labelled goat anti-human IgA (Abcam, Cambridge, UK) diluted 1:5000 in ELISA diluent were added and plates were incubated at 37 °C for 30 min. 100 μL/well of tetramethylbenzidine (KPL, Gaithersburg, MD) was added and plates were incubated at RT for five minutes before 100 μL/well of 0.5 M H_2_S0_4_ was added to stop assay development. Absorbance was read at 450/620 nm. 

C-reactive protein (CRP) [16] was measured using human CRP ELISA kit (eBioscience, San Diego, CA, USA) (analytical sensitivity of 150 pg/mL) with duplicate samples and standards according to manufacturers’ protocols. CRP levels >6 μg/mL was considered above normal. 

All samples were tested with blinding of PTB status.

### 2.3. Statistical Analysis

Average optical density (OD) and variance of duplicates were calculated in Microsoft Excel 2011 (Microsoft, Redmond, WA, USA). All sample ODs had less than 10% variance and were included in the analysis. Concentration of CRP was interpolated from OD and standards by generating a four-parameter logistic curve-fit in GraphPad Prism-7 (GraphPad Software, La Jolla, CA, USA). The following statistical tests were conducted in Stata-11 (StataCorp LP, College Station, TX, USA): (1) Kruskal-Wallis and Pearson’s chi-square tests for comparisons of numerical and binary/categorical variables, respectively between diagnostic groups (SPCP, SNCP and non-TB patients), followed by multinomial logistic regression to identify significant predictors and relative risk (RR); (2) analysis of variance (ANOVA) and multivariate ANOVA (MANOVA) for comparison of biomarker distribution and interaction across diagnostic groups, HIV status and country; and Pearson’s test for correlations; (3) Iterative multiple regression to identify significant host predictors of biomarkers by diagnostic group and HIV status and beta coefficients; and (4) area under receiver operating characteristics (AUC) for analyzing diagnostic values of biomarkers to discriminate PTB patients against non-TB patients, and according to HIV and smear status. Data was natural log transformed whenever necessary to meet requirements of variance homogeneity for ANOVA and regression analysis and/or a more conservative alpha value (*p* < 0.01) was used to determine significance. Otherwise, *p* < 0.05 was considered significant, with Bonferroni adjustments for multiple comparisons.

## 3. Results

### 3.1. Host Factors with Diagnostic Status and Biomarker Distribution

A significantly higher proportion of SPCP and SNCP patients were male and had abnormal chest x-ray (CXR) compared to non-TB patients. A higher proportion of SPCP smoked, had cavitary lesions on CXR, and reported coughing for five weeks or longer compared to SNCP and non-TB patients. A higher proportion of SNCP had wasting compared to SPCP and non-TB patients (Table 1). Subsequent multinomial logistic regression revealed that SPCP were more likely male (RR: 2.9, 95% CI: 1.3–6.5), had longer duration of coughing (RR: 3.0, 95% CI: 1.4–6.3), and were less likely to be older (RR: 0.2, 95% CI: 0.1–0.5) compared to non-TB. SNCP were more likely to be male (RR: 3.1, 95% CI: 1.4–7.0), and less likely to be older (RR: 0.4, 95% CI: 0.1–0.9) compared to non-TB. SPCP were more likely to have longer duration of coughing compared to SNCP (RR: 3.3 95% CI: 1.2–8.7). 

Multiple multinomial regressions were conducted to analyse host factor relationship with biomarker distribution. Global regression with all host factors followed by iterative removal of predictors with *p* < 0.25, adjusted for diagnostic group, resulted in final models that identified: (1) expectoration, chest pain, wasting, higher culture count association with higher CRP concentration (F(13, 203) = 9.45, *p* < 0.001); (2) higher smear count association with higher anti-A60 IgG (F(7, 359) = 1.78, *p* = 0.090); and (3) expectoration and enlarged lymph nodes association with lower anti-A60 IgA (F(9, 158) = 2.31, *p* = 0.018) (Table 2).

Finally, SPCP, SNCP and non-TB patients were analysed separately to determine whether host factor-biomarker relationships varied by diagnostic group. Among non-TB, CRP was higher in patients with wasting and HIV+ve, while patients who consumed alcohol had higher anti-A60 IgG. In SPCP, patients with weight loss, Bacillus Calmette–Guérin (BCG) vaccination and HIV had higher CRP while patients with weight loss had higher anti-A60 IgG. An abnormal CXR was associated with lower anti-A60 IgG in SPCP. In SNCP, fever and high alcohol consumption was positively associated with CRP while BCG vaccination and high alcohol consumption was positively associated with anti-A60 IgA. Higher culture count appeared to be negatively associated with anti-A60 IgA in SNCP (Table 2).

### 3.2. Distribution and Correlation of Seroimmunological Biomarkers 

SPCP, SNCP and non-TB appeared to have comparable levels of anti-A60 IgG (*p* = 0.126) and anti-A60 IgA. CRP was significantly higher in SPCP compared to SNCP (*p* < 0.001) and non-TB patients (*p* < 0.001) (Figure 1A). When data were analyzed by HIV status (Figure 1B), anti-A60 IgG remained similar across diagnostic groups, while anti-A60 IgA was lower in SPCP compared to non-TB (*p* = 0.036) in HIV+ve patients. Furthermore, a significant interaction was observed between TB diagnosis and HIV status (*p* = 0.003) in anti-A60 IgA. CRP was significantly higher in SPCP (*p* < 0.001) and SNCP (*p* < 0.05) compared to non-TB in both HIV−ve and HIV+ve patients, while a significant interaction between TB diagnosis and HIV status (*p* = 0.005) was also observed.

In further analysis by country (data not shown), anti-A60 IgG was not significantly different across groups in Vietnamese and South African patients. Anti-A60 IgA in SPCP was higher than SNCP (*p* = 0.021) and non-TB (*p* = 0.004) in HIV−ve South African patients; however anti-A60 IgA was lower in SPCP than SNCP and non-TB in HIV+ve patients—but this was not statistically significant. CRP was higher in SPCP and SNCP compared to non-TB in HIV−ve (*p* = 0.021) and HIV+ve South African patients (*p* = 0.026). Significant interaction was observed between country and HIV status (*p* = 0.007), and between TB status and HIV status (*p* = 0.005) (but not between TB status and country) for CRP.

The influence of HIV status on heterogeneity of seroimmunological response was further visualized by comparing biomarker reactivity across individual patients in SPCP, SNCP and non-TB groups (Figure 2A). Although non-TB patients generally had lower reactivity (green shade) there were considerable numbers of patients with higher reactivity (yellow-red shade). Similarly, the active PTB patients had many patients with low reactivity, which resulted in significant overlaps between these diagnostics groups and underscores difficulty in discriminating these heterogeneous populations using serological biomarkers. The heterogeneity differed by HIV status, and as shown in scatterplots (Figure 1), lower anti-A60 IgG reactivity in SNCP versus non-TB for HIV+ve and lower anti-A60 IgA in SPCP versus non-TB in HIV+ve was observed. 

Pearsons test and scatterplots (Figure 2B–D) showed significant positive correlation between anti-A60 IgA and IgG, in non-TB HIV−ve (r = 0.418, *p* < 0.001) and HIV+ve patients (r = 0.498, *p* < 0.001); in SPCP HIV−ve (r = 0.451, *p* < 0.001) and HIV+ve (r = 0.458, *p* = 0.003). Conversely, CRP did not correlate with anti-A60 IgA or anti-A60 IgG across all groups. 

### 3.3. Biomarker Diagnostic Evaluation

Figure 3 illustrates the variability in diagnostic accuracy across different PTB subpopulations, and across different combinations of biomarkers assessed for diagnostic potential. Individually, anti-A60 IgG, anti-A60 IgA and CRP had limited diagnostic accuracy, although CRP had significantly higher accuracy (*p* < 0.001) compared to the antibody markers. The accuracy of combinations containing CRP appeared to follow the accuracy of CRP, and the highest accuracy was generally achieved using combinations of anti-A60 IgG and CRP, particularly in SPCP vs. non-TB who are HIV−ve (AUC: 0.838, 95% CI: 0.783–0.894). Conversely, anti-A60 IgA combined with CRP had highest accuracy in diagnosing HIV+ve populations of PTB vs. Non-TB (AUC: 0.687, 95% CI: 0.598–0.777) and SPCP vs. non-TB (AUC: 0.774, 95% CI: 0.682–0.865). The lowest AUC observed across study subpopulations was using a combination of anti-A60 IgG and IgA (without CRP). 

## 4. Discussion

The primary aim of this analysis was to determine the influence of HIV co-infection and host factors on serological biomarkers, using anti-A60 IgG and IgA and C-reactive protein as MTB-specific and innate markers of host immunity, respectively. The data however has highlighted the high degree of variation and complexity in relationships between biomarker distributions and host factors (including HIV status), which have been previously implied but not comprehensively evaluated by preceding studies [15,28,29]. Of note, the lower level of anti-A60 IgA in SPCP compared to non-TB has not been reported elsewhere. While HIV+ve individuals reportedly displayed higher infection enhancement in IgA response to HIV antigens [30], it is unclear how this may relate to altered IgA binding against TB antigens during TB-HIV co-infection. Visually, the pattern of distribution for CRP appeared consistent across study subpopulations (Figure 1). However, significant interaction observed between country and HIV for CRP as well as diagnostic status and HIV for anti-A60 IgA and CRP suggests that the distribution of these biomarkers across countries and diagnostic groups may be influenced by the distribution of HIV+ve subpopulations. The low correlation between CRP and anti-A60 antibodies provides a rationale for combining an inflammation marker with an MTB-specific marker to better capture the heterogeneity of PTB patient immunity [23,31]. However, unlike previous studies reporting significant increase in diagnostic value for discriminating SPCP from non-TB HIV−ve populations when a variety of antigens and antibody classes were combined [22], in this sample population different biomarker combinations only marginally improved on CRP alone. In general, sensitivity and specificity of the assays individually and in combination, were generally low, as has been reported in the literature [13,15], but were highest in discriminating SPCP from non-TB in HIV−ve subpopulations, which is in line with proteomic biomarker studies reported previously [32,33].

The relationship between host characteristics and diagnostic status observed here may reflect the sampling bias from passive case detection, as patients who present to clinics tend to be symptomatic [34]. However, lower cough duration and chest x-ray abnormality in SNCP compared to SPCP follows a previous observation that the absence of cough in the presence of other symptoms is characteristic of smear-negative PTB [35]. Wasting, defined as BMI of <18.5, is a well-known symptom of TB arising from loss of lean and fat tissue [36], while smear-negative PTB, implying a lower AFB burden, is often observed in HIV+ve patients [37,38]. Hence, the higher proportion of wasting in SPCP compared to SNCP identified in univariate analysis was no longer observed in multivariate analysis controlled for HIV status. 

A significant association between symptoms characteristic of PTB such as expectoration, wasting and chest pain with higher CRP observed in this study appears to be consistent with recent literature [39]. The significant positive association between culture count and smear count, with CRP and anti-A60 IgG, respectively, follows postulations that AFB burden influences seroimmunological heterogeneity [19,20,39]. The inverse relationship observed between enlarged lymph nodes and anti-A60 IgA may relate to previous implications of IgA deficiency being a cause of idiopathic lymphadenapathy [40], whereby patients with less propensity to produce IgA may be more likely to manifest lymphadenapathy. Conversely, lower anti-A60 IgA observed in patients with expectoration appears to contradict a recent report of increased anti-tuberculous glycolipid IgA in patients with cavitary TB and bronchoiectasis [41]—but this may reflect the difference in antigens evaluated. Among the demographic host factors evaluated, only alcohol consumption had significant positive association with CRP, anti-A60 IgG and anti-A60 IgA, though the degree of association varied by PTB subpopulation. Excess alcohol use is associated with greater manifestations of PTB such as cavitary lesions and AFB burden [42], and reported to increase the expression of genes associated with lung inflammation [43]. However, to our knowledge, this is the first report of measurable association between alcohol consumption and serodiagnostic biomarkers such as CRP, and anti-A60 antibodies. 

While the present findings may enrich the literature on host factor-biomarker relationships, the sheer complexity of these relationships reiterates the complexity and heterogeneity in tuberculosis pathology and host-pathogen interactions [44], which continues to thwart coherent understanding of the patient immunological response in TB infection. Importantly, these data strongly imply limited application of host factor screening in TB serodiagnostic algorithms. Indeed, the only consistent pattern is the significant decline in accuracy of different serodiagnostic algorithms in discriminating active TB from non-TB among individuals with smear-negative TB and HIV co-infection (or both). Serodiagnostic development based on these data appears to preclude use in many individuals living in areas with high proportion of HIV co-infection (and hence SNCP), such as in sub-Saharan Africa. Further serodiagnostic research and development may only be able to target use in settings with high TB but traditionally low HIV prevalence in the general population, such as in the Asia Pacific [1]. 

Finally, the strengths of this study include: relatively large number of samples including HIV+ve and smear-negative subpopulations, samples were tested in a blinded manner, host marker of innate immunity and TB-specific antibodies were assessed, and multivariate analyses were employed. Limitations of this study include: cross-sectional nature of sampling, the lack of healthy controls and pediatric samples, and lack of patient vitamin D status, which may also be a risk factor of TB disease [45]. Furthermore, the status of latent TB was not determined in this study population. This is an important caveat as there have been suggestions that false positive biomarker-based results among non-TB patients in higher burden countries may stem from subclinical manifestations of latently infected individuals [32,46].

## 5. Conclusions

TB serological biomarker discovery and development have long been hampered by a lack of optimal biomarkers and a heterogeneous patient immune response. The ideal biomarker is targeted to perform consistently regardless of host factors and co-morbidities. The data presented suggest that several host factors including alcohol consumption and factors related to disease symptoms and AFB burden play an important role in biomarker distribution. Based on this analysis, it appears that seroimmunological biomarkers do not fit the current target product profile and have limited potential for improving case detection in HIV+ve and SNCP populations. Until there is a revolutionary shift in TB biomarker discovery, the marked difference in immune response between HIV+ve and HIV−ve individuals, as well as complex relationships between a variety of host and bacterial factors with seroimmunological response, suggest the need of evaluating different diagnostic strategies, including specific target biomarker characteristics, to separately diagnose these patient subpopulations. 

## Figures and Tables

**Figure 1 tropicalmed-04-00057-f001:**
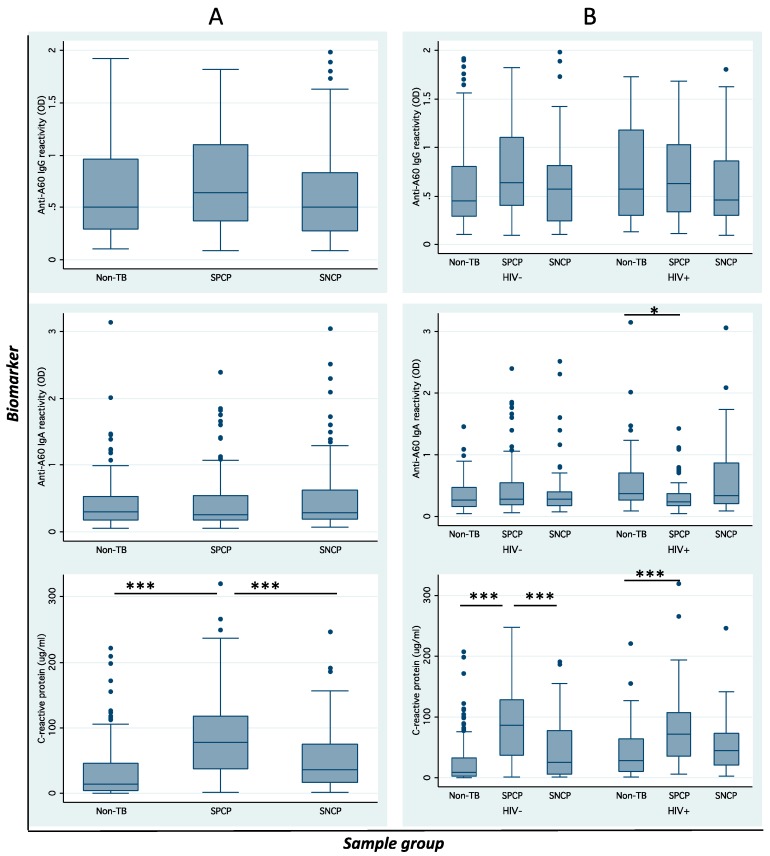
Biomarker distribution. Boxplots of anti-A60 immunoglobulin G (IgG), anti-A60 immunoglobulin A (IgA) and C-reactive protein (CRP) across diagnostic groups (**A**) without HIV status and (**B**) with HIV status taken into account demonstrates influence of HIV on biomarker distribution. Error bars indicate standard deviation. Asterisks indicate statistical significance of *p* < 0.05 (*), *p* < 0.01 (**) and *p* < 0.001 (***). CRP levels >6 μg/mL was considered above normal.

**Figure 2 tropicalmed-04-00057-f002:**
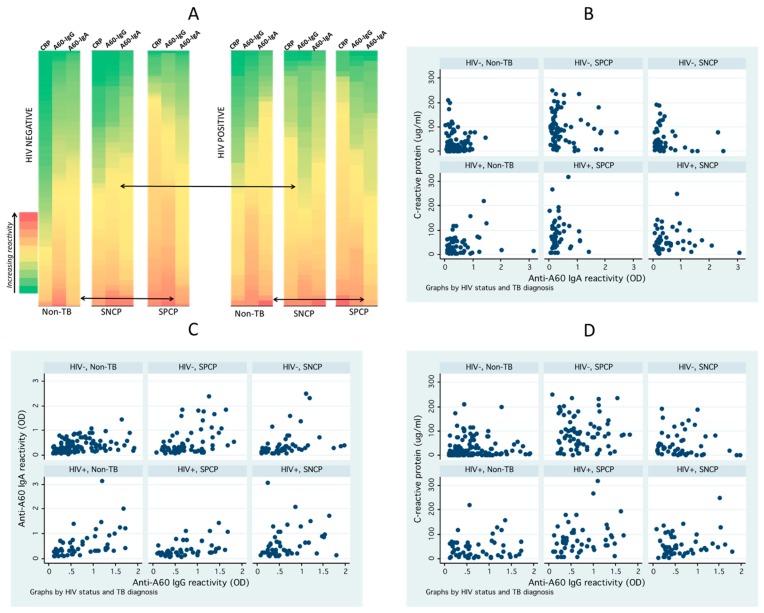
Biomarker reactivity and correlation by diagnostic group and HIV status. (**A**) Heat map of biomarker reactivity illustrates high degree of heterogeneity across patients and difference in reactivity between HIV−ve and HIV+ve patients. Individual patients are colored green to red according to increasing relative reactivity/concentration for biomarker (CRP, anti-A60 IgG, anti-A60 IgA). (**B**–**D**) Scatterplots of biomarkers shows slight but significant correlation of anti-A60 antibodies, and lack of correlation between anti-A60 antibodies and CRP.

**Figure 3 tropicalmed-04-00057-f003:**
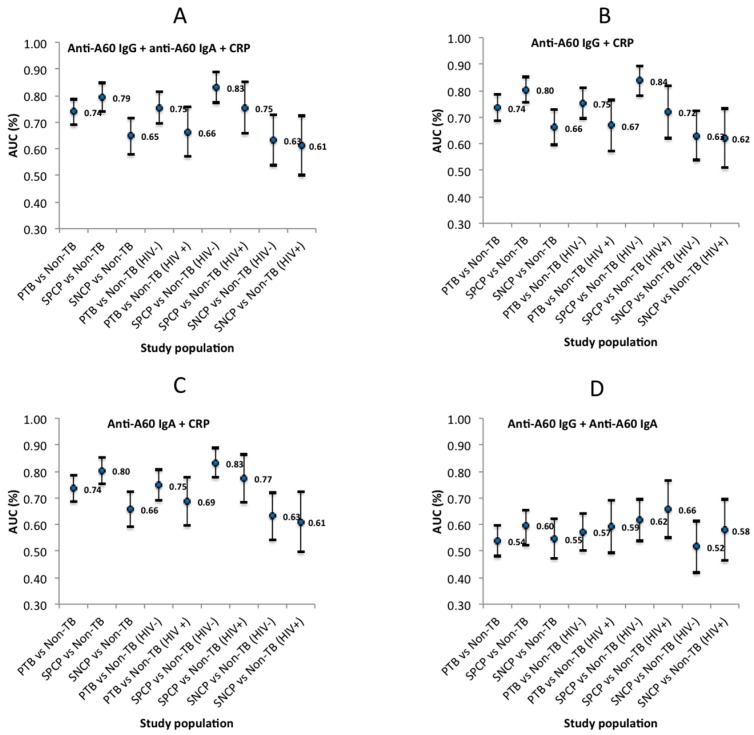
Estimated area under receiver operating characteristics (AUC) (with 95% CI) of four biomarker combinations; (**A**) anti-A60 IgG + anti-A60 IgA + CRP, (**B**) anti-A60 IgG + CRP (**C**) anti-A60 IgA + CRP, and (**D**) anti-A60 IgG + anti-A60 IgA for different study subpopulations.

**Table 1 tropicalmed-04-00057-t001:** Demographic and clinical characteristics of the study population.

Characteristic Frequency (%)	Non-TB (*n* = 182)	SPCP (*n* = 112)	SNCP (*n* = 100)	*p* Value
Age group				0.016
18–24 years	22 (14)	35 (31)	19 (19)	
25–35 years	43 (28)	25 (22)	24 (24)	
36–45 years	39 (25)	29 (26)	32 (32)	
46 years and older	52 (33)	23 (21)	25 (25)	
Age in years: Mean (SD)	40 (15)	34 (11)	37 (37)	0.005
Male	79 (51)	87 (74)	68 (68)	<0.001 *
Cigarette Smoker	42 (27)	61 (54)	35 (35)	<0.001 *
Country **				
Viet Nam	72 (40)	71 (58)	40 (40)	
South Africa	110 (60)	51 (42)	60 (60)	
Alcohol consumption				0.008
No	88 (72)	50 (47)	51 (61)	
Less	25 (20)	41 (38)	19 (23)	
Weekly	6 (5)	10 (9)	12 (14)	
Daily	3 (2)	6 (6)	2 (2)	
Frequent to intoxication	2 (2)	2 (2)	1 (1)	
HIV positive **	50 (27)	49 (40)	50 (50)	
Clinical symptoms and Medical History				
Cough duration				<0.001 *
1–4 weeks	97 (78)	52 (48)	55 (71)	
5–10 weeks	17 (14)	33 (30)	15 (19)	
11 weeks or longer	10 (8)	24 (22)	7 (9)	
Cough duration (weeks): Mean (SD)	7 (28)	10 (18)	5 (5)	<0.001 *
Expectoration	93 (98)	66 (97)	51 (94)	0.512
Haemoptysis	19 (12)	16 (14)	15 (15)	0.773
Dsypnea	62 (40)	44 (39)	53 (53)	0.068
Night sweats	88 (56)	70 (63)	68 (69)	0.141
Weight loss	95 (61)	78 (70)	70 (71)	0.178
Fever above 38 °C	76 (49)	74 (66)	67 (68)	0.002 *
Chest pain	79 (51)	65 (58)	66 (67)	0.041
Malaise	95 (61)	73 (65)	70 (71)	0.277
Wasting	12 (8)	21 (19)	28 (28)	<0.001 *
Enlarged lymph nodes	1 (1)	5 (5)	5 (6)	0.107
Abnormal chest x-ray	96 (64)	116 (98)	83 (92)	<0.001 *
Chest x-ray interpretation				0.011
Infiltrate or consolidated	70 (76)	74 (65)	52 (63)	
Pleural effusion	1 (1)	2 (2)	4 (5)	
Cavitary lesion	2 (2)	20 (18)	9 (11)	
Tuberculoma	0 (0)	0 (0)	1 (1)	
Mediastinal/hilar lymphoadenopathy	5 (5)	3 (3)	5 (6)	
Micronodules (Miliar)	0 (0)	5 (4)	3 (4)	
Other	14 (15)	9 (8)	8 (10)	
Chest x-ray conclusion				<0.001 *
TB likely	15 (16)	114 (98)	64 (77)	
Pneumonia or atypical TB	35 (36)	1 (1)	16 (19)	
Pneumonia (TB unlikely)	27 (28)	0 (0)	0 (0)	
Other	19 (20)	1 (1)	3 (4)	
Contact with active TB case	27 (17)	21 (19)	27 (27)	0.142
BCG vaccinated	81 (63)	63 (67)	62 (67)	0.757
Previous TB	36 (23)	13 (11)	21 (22)	0.029
General appearance				0.004
Not ill	41 (26)	13 (12)	13 (13)	
Mildly ill	68 (44)	47 (42)	51 (51)	
Moderately/gravely ill	47 (30)	52 (46)	36 (36)	

Note: Patient diagnosis confirmed with culture (C) and sputum smear microscopy (SS) with follow-up at two months. Statistical comparisons were conducted using Kruskal-Wallis for numerical variables and Pearson chi-square tests for binomial/categorical variables, two-tailed. * *p* values < 0.002 (Bonferonni adjusted) are significant. ** Human immunodeficiency virus (HIV) and country were pre-determined during sample selection, hence not statistically compared.

**Table 2 tropicalmed-04-00057-t002:** Regressions to predict biomarkers with host factors.

Study Population	Biomarker (n)	Model Predictors	F Statistic	Significant Predictors	β Coefficient	LL	UL	*p* Value
Total	CRP (217)	Expectoration, chestpain, wasting, smear count, culture count, TB status	F(13, 203) = 9.45	Expectoration	3.39	1.23	9.39	0.019
				Chest pain	1.55	1.07	2.23	0.019
				Wasting	3.42	1.90	6.23	<0.001
				Culture count	2.92	1.31	6.49	0.009
	anti-A60 IgG (367)	Night sweats, smear count, TB status	F(9, 158) = 2.31	Smear count	0.46	0.06	0.86	0.023
	anti-A60 IgA (168)	Expectoration, Enlarged lymph node, HIV status, smear count, TB status	F(9, 158) = 2.31	Expectoration	−1.12	−1.77	−0.49	0.001
				Enlarged lymph node	−0.68	−1.2	−0.16	0.011
Non-TB	CRP (123)	Enlarged lymphnode, wasting, HIV status	F(3, 119) = 5.87	Wasting	3.82	1.30	11.25	0.015
				HIV	1.86	1.06	3.25	0.03
	Anti-A60 IgG (124)	Malaise, Alcohol consumption		Alcohol consumption	0.9	0.32	1.49	0.003
SPCP	CRP (92)	Weight loss, BCG, HIV status, Alcohol consumption	F(7,84) = 2.95	Weight loss	1.84	1.17	2.89	0.009
				BCG	1.62	1.11	2.34	0.013
				HIV	0.63	0.44	0.90	0.014
	Anti-A60 IgG (109)	Weight loss, Abnormal chest x-ray, HIV status, Alcohol consumption	F(5, 118) = 3.00	Weight loss	0.28	0.09	0.48	0.004
				Abnormal chest x-ray	−0.68	−1.29	−0.07	0.03
SNCP	CRP (85)	Fever, Alcohol consumption, Culture count	F(9, 75) = 2.92	Fever	2.89	1.62	5.21	0.001
				Alcohol consumption	10.18	1.01	102.51	0.049
	Anti-A60 IgA (71)	Sex, malaise, TB history, BCG, Smoking, Abnormal chest x-ray, Age, Culture count, Alcohol consumption	F(17, 53) = 2.85	BCG	0.44	0.14	0.73	0.004
				Alcohol consumption	1.8	0.59	3	0.004
				Culture count	−0.57	−0.98	−0.15	0.009

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
