# Peer review of "An Observational Case-Control Study to Determine Human Immunodeficiency Virus and Host Factor Influence on Biomarker Distribution and Serodiagnostic Potential in Adult Pulmonary Tuberculosis"

_tropicalmed, 2019, doi:10.3390/tropicalmed4020057_

Round 1
Reviewer 1 Report
A60 antigen studies in TB have been reported since 1997.
Strengths: The authors have now used an extensive bank of well defined serological specimens from two different settings; they included large numbers of the relevant subgroups for robust comparisons of smear positive and smear negative culture positive PTB patients, either or not HIV co-infected; and a controle group comprising both HIV+ and HIV-ve individuals.
My understanding when reading is, that all of the study participants were enrolled because they were suspected to have TB; is that right? this would be an enormous strength of the study, as the test algorithm is highly relevant: it is among the TB suspects that a serodiagnostic test algorithm should be used and therefore first tested.
They describe the procedures adequately and their data analysis is robust and extensive
The illustrations are data-rich and help to understand the different findings
Weaknesses include those already mentioned by the authors, though I feel a lack of vit D and latent TB is not what bothers me at all; the cross sectional nature is somehow worrying as who knows, some of these individuals may potentially be mis-classified in retrospect had the study teams followed these individuals over time; my major concern is that we do not know what the alternative diagnoses were in the patients classified with HIV+ or HIV- non-TB. Did they have bronchiectasis? aspiration pneumonia? community-acquired pneumonia? COPD? lung cancer? streptococcal or Gram-negative pleural empyema? PcP? It seems they were less sick than the SPCP TB patients, so my guess is they bronchiectasis and COPD may have been the usual alternative diagnosis . .
Finally: careful re-reading and teasing out of typographical error might help to improve the paper. My final advise would be to slightly shorten the lengthy text. The Discussion could be more to-the-point, without elaborating on positive findings; the bottom-line s rather that the current contribution of this test panel does not help to meet the goals so clearly spelled out and defined in the elegantly written Introduction.
Author Response
Reviewer 1 Comments
A60 antigen studies in TB have been reported since 1997.
Point 1
Strengths: The authors have now used an extensive bank of well defined serological specimens from two different settings; they included large numbers of the relevant subgroups for robust comparisons of smear positive and smear negative culture positive PTB patients, either or not HIV co-infected; and a controle group comprising both HIV+ and HIV-ve individuals.
My understanding when reading is, that all of the study participants were enrolled because they were suspected to have TB; is that right? this would be an enormous strength of the study, as the test algorithm is highly relevant: it is among the TB suspects that a serodiagnostic test algorithm should be used and therefore first tested.
Response 1
We are pleased the reviewer acknowledges and understands the significance of the study population used in this study. The reviewer is correct in understanding that the sera samples used in this study were obtained from participants who were enrolled due to a provisional diagnosis of tuberculosis, but were then excluded after two consecutive microbiological evaluations (sputum smear microscopy and culture). These study groups represent a more real-world scenario where serodiagnostic algorithms are more likely to be applied, rather than comparison with healthy volunteer controls.
Point 2
They describe the procedures adequately and their data analysis is robust and extensive. The illustrations are data-rich and help to understand the different findings
Response 2
We thank the reviewer for this positive comment, and are pleased that the analysis was deemed appropriate.
Point 3
Weaknesses include those already mentioned by the authors, though I feel a lack of vit D and latent TB is not what bothers me at all; the cross sectional nature is somehow worrying as who knows, some of these individuals may potentially be mis-classified in retrospect had the study teams followed these individuals over time; my major concern is that we do not know what the alternative diagnoses were in the patients classified with HIV+ or HIV- non-TB. Did they have bronchiectasis? aspiration pneumonia? community-acquired pneumonia? COPD? lung cancer? streptococcal or Gram-negative pleural empyema? PcP? It seems they were less sick than the SPCP TB patients, so my guess is they bronchiectasis and COPD may have been the usual alternative diagnosis.
Response 3
While FIND has given reasonable effort to follow-up and re-examine sputum smear and culture two months after initial non-TB diagnosis, we agree that misdiagnosis of non-TB patients is a valid concern that cannot be sufficiently ruled out given the increasing spectrum and timeline of TB disease. This perhaps raises the question of the use of smear and sputum culture as the gold standard for diagnosis and whether molecular methods are required to truly rule out TB for biomarker research cohorts, but this may be beyond the scope of this article to discuss. We acknowledge in the methods that extrapulmonary and latent TB could not be excluded from this group, but to address the reviewer’s concern regarding the status of non-TB patients based on available information of chest x-ray conclusions, we can additionally report in the revision in Ln 111-115:
“Available chest x-ray conclusions for 82 non-TB patients (45%) indicate that 46 non-TB patients were diagnosed with “pneumonia or atypical TB”, seven were diagnosed as “TB likely”, two patients were diagnosed with “COPD” (chronic obstructive pulmonary disorder), and one patient was diagnosed with “PCP” (Pneumocystis pneumonia), while nine patients were noted to have “evidence of previous TB”.”
Point 4
Finally: careful re-reading and teasing out of typographical error might help to improve the paper. My final advise would be to slightly shorten the lengthy text. The Discussion could be more to-the-point, without elaborating on positive findings; the bottom-line s rather that the current contribution of this test panel does not help to meet the goals so clearly spelled out and defined in the elegantly written Introduction.
Response 4
We have revised the manuscript to correct typographical errors directly in the revised manuscript. However, we believe that given the complexity of the relationship between the predictors and antibody/CRP response, the discussion is necessary and may be useful should other researchers revisit the use of host predictors of immunological responses for TB or similar diseases.
Reviewer 2 Report
The definition of diagnostic biomarkers for tuberculosis is an important research priority, as existent diagnostic tools are insensitive (smear), or slow (culture) or expensive (Xpert). The study used sera from well clinically defined populations to evaluate three biomarkers. Although a case-control design is not ideal for biomarker evaluation, it is appropriate for exploratory studies. Blinding was used. Some specimens were excluded based on more than 10% variance, and the number of these exclusions should be included in the paper. Analysis is appropriate, with analysis also done on predictors among non-TB controls.
The influence of host factors and burden of bacilli on biomarkers is complex, and is described well here. The TB patients who need a new diagnostic test the most (HIV+, smear negative) are the least helped by current biomarker assays, unfortunately.
The paper is very well written and the conclusions are sound. I think the paper should be published with one revision (description of excluded specimens).
Author Response
Reviewer 2 Comment
The definition of diagnostic biomarkers for tuberculosis is an important research priority, as existent diagnostic tools are insensitive (smear), or slow (culture) or expensive (Xpert). The study used sera from well clinically defined populations to evaluate three biomarkers. Although a case-control design is not ideal for biomarker evaluation, it is appropriate for exploratory studies. Blinding was used. Some specimens were excluded based on more than 10% variance, and the number of these exclusions should be included in the paper. Analysis is appropriate, with analysis also done on predictors among non-TB controls.
The influence of host factors and burden of bacilli on biomarkers is complex, and is described well here. The TB patients who need a new diagnostic test the most (HIV+, smear negative) are the least helped by current biomarker assays, unfortunately.
The paper is very well written and the conclusions are sound. I think the paper should be published with one revision (description of excluded specimens).
Response
We thank the reviewer for a concise and fair assessment of the manuscript. Although we described in the methodology that only specimens with average antibody reactivity with less than 10% variance were included in the analysis, there were actually no specimens with more than 10% OD variance and hence no samples from this study population were dropped from the analysis. We have clarified this in the methods by revising the sentence to read in Ln 145:
“All sample ODs had less than 10% variance and were included in the analysis.”